# Cardiovascular Tropism and Sequelae of SARS-CoV-2 Infection

**DOI:** 10.3390/v14061137

**Published:** 2022-05-25

**Authors:** Oleksandr Dmytrenko, Kory J. Lavine

**Affiliations:** 1Center for Cardiovascular Research, Departmental of Medicine, Cardiovascular Division, Washington University School of Medicine, 660 South Euclid Campus Box 8086, St. Louis, MO 63110, USA; oleksandr.dmytrenko@wustl.edu; 2Department of Pathology and Immunology, Washington University School of Medicine, St. Louis, MO 63110, USA; 3Department of Developmental Biology, Washington University School of Medicine, St. Louis, MO 63110, USA

**Keywords:** SARS-CoV-2, COVID-19, post-acute sequelae of SARS-CoV-2 infection, cardiomyocyte, pericyte, myocarditis, thrombosis

## Abstract

The extrapulmonary manifestation of coronavirus disease-19 (COVID-19), caused by severe acute respiratory syndrome coronavirus 2 (SARS-CoV-2), became apparent early in the ongoing pandemic. It is now recognized that cells of the cardiovascular system are targets of SARS-CoV-2 infection and associated disease pathogenesis. While some details are emerging, much remains to be understood pertaining to the mechanistic basis by which SARS-CoV-2 contributes to acute and chronic manifestations of COVID-19. This knowledge has the potential to improve clinical management for the growing populations of patients impacted by COVID-19. Here, we review the epidemiology and pathophysiology of cardiovascular sequelae of COVID-19 and outline proposed disease mechanisms, including direct SARS-CoV-2 infection of major cardiovascular cell types and pathogenic effects of non-infectious viral particles and elicited inflammatory mediators. Finally, we identify the major outstanding questions in cardiovascular COVID-19 research.

## 1. Epidemiology of Cardiovascular COVID-19 Manifestations

The severe acute respiratory syndrome coronavirus 2 (SARS-CoV-2) pandemic has profoundly impacted global health, health care delivery systems, and the economy, leading to more than 484 million cases and 6.1 million deaths as of 1 April 2022 [1]. Unfortunately, the emergence of SARS-CoV-2 variants of concern (VOC) has prolonged and intensified the current pandemic [2,3]. The ensuing illness, termed coronavirus disease-19 (COVID-19), predominantly manifests as a respiratory disease, although extrapulmonary involvement occurs in a sizable number of patients [4,5]. Systemic manifestations of COVID-19 have been noted across organ systems with predominate involvement of the cardiovascular, renal, gastrointestinal, and central nervous systems [2,3,4,5].

Patients infected with SARS-CoV-2 display a wide range of disease severity, ranging from asymptomatic or mild infection to critical illness with multiple organ failure [6,7]. Critically ill cases of COVID-19 present with progressive multiorgan system failure and cardiovascular collapse, often requiring inotropic, vasopressor, and/or mechanical support, and displaying high mortality rates. Disproportionate rates of critical illness are observed in vulnerable populations, including the elderly, those with underlying cardiovascular comorbidities, and underrepresented minorities [8,9,10,11,12]. Throughout the pandemic, pre-existing cardiovascular disease has continued to represent an important risk factor for COVID-19 critical illness and mortality [2]. 

Cardiac complications of COVID-19 occur in 20–44% of acutely hospitalized patients, and constitute an independent risk factor for COVID-19 mortality [8,13,14,15]. Factors such as patient age, the severity of COVID-19 pneumonia, preexistent cardiac disease, immunocompromised state, and the use of cardiotoxic therapies predispose these patients to cardiovascular complications [16,17]. During acute infection, cardiac manifestations include myocardial injury (elevated serum troponin levels), myocarditis, pericarditis, heart failure, acute coronary syndromes, and arrhythmias (Figure 1) [18,19,20,21,22,23,24,25,26,27]. Among those patients presenting with chest pain and/or heart failure, cardiac magnetic resonance imaging has revealed signs of myocardial and pericardial inflammation (delayed contrast enhancement, T1 mapping, T2 signal) [28,29]. Surprisingly, MRI evidence of cardiac inflammation was reported in patients who had seemingly recovered, further highlighting an underappreciated cardiac component of this disease, even in its milder forms [28]. 

Vascular involvement is particularly evident and often presents as acute venous (deep-vein thrombosis and pulmonary embolism) and arterial (stroke and critical limb ischemia) thrombosis (Figure 1) [30,31,32,33,34]. Severely ill patients present with coagulopathy and disseminated intravascular coagulation, while bleeding remains uncommon [35]. Among laboratory findings, elevated D-dimer, fibrin degradation products, and fibrinogen were observed among the patients with the most severe form of the disease [36]. The current in-hospital therapies for COVID-19 patients include thrombosis prophylaxis with heparin or factor Xa inhibitors [37,38].

Long-term sequelae of COVID-19 infection are now being identified in patients who remain symptomatic beyond the acute phase. These manifestations have been termed post-acute sequelae of SARS-CoV-2 infection (PASC) by the National Institute of Health [39]. Several case reports and studies have emerged describing post-acute COVID-19 symptoms in the weeks following the initial infection, with significant symptoms observed in young and otherwise healthy individuals who had mild acute COVID-19 symptoms [39,40]. A wide array of cardiovascular symptoms have been described in this population, such as fatigue, exertional dyspnea, chest pain, and palpitations [41]. Remarkably, only 30% of patients fully recovered 6 months after infection [42]. A recent study of 150,000 patients revealed that a history of acute COVID-19 predisposed patients to a higher risk of adverse cardiovascular events, such as arrhythmias, stroke, and heart failure, which occurred over a 12-month period following the COVID-19 diagnosis. The risk of these events correlated with the severity of the acute COVID-19, but was significant even for those patients with mild disease and limited prior risk factors for cardiovascular disease [43]. While much remains to be learned regarding the epidemiology and natural history of cardiovascular PASC, these early observations signify that cardiovascular damage is a long-lasting feature of COVID-19 in select individuals.

Highly effective vaccines against the SARS-CoV-2 spike protein have changed the course of the COVID-19 pandemic and provided significant protection from severe disease [44,45]. The successful implementation of these vaccines may reduce the cardiovascular sequalae by limiting infection and the overall burden of disease. It remains to be elucidated whether breakthrough infections in vaccinated individuals, characteristic of multiple variants of the virus [46,47], will lead to increased cardiovascular risk. Rare cardiovascular complications of COVID-19 vaccination are myocarditis and pericarditis, observed with mRNA-based vaccines [48,49,50,51]. The incidence of these complications was highest in the 12–17 age group of males after the second dose of vaccine (22–36 cases per 100,000). It is important to note that the risk of myocarditis from SARS-CoV-2 infection is 2–7 times higher than that from vaccination for this age group. Thus, vaccination remains the recommended preventative strategy for cardiac complications of COVID-19 [52]. 

## 2. Pathophysiology

The underlying cause(s) of the cardiovascular manifestations of acute COVID-19 and PASC remain a topic of considerable debate. Direct viral infection of cardiovascular cell types, systemic inflammation, and microvascular thrombosis have each been implicated in the pathogenesis of acute COVID-19 (Figure 2). Patients with acute COVID-19 display numerous systemic derangements, including marked increases in circulating inflammatory mediators [53,54], activation of the complement cascade [55], impaired fibrinolysis [56], platelet activation and aggregation [57]. While associations between systemic inflammation and acute cardiovascular sequelae of COVID-19 exist [58], a causative relationship is yet to be rigorously established. Much less is known regarding the pathology of PACS. Cardiac MRI findings compatible with myocarditis were only observed in a subset of individuals [59], and increasing evidence may point towards the involvement of the vasculature [60] and autonomic nervous systems [61,62]. 

Postmortem analysis of patients who succumbed to acute COVID-19 has revealed evidence of cardiac involvement without clinically apparent features of heart failure or myocarditis. Myocardial necrosis, myocarditis, and microthrombi in capillaries, arterioles, and small arteries were apparent in approximately 35% of cases. Abundant interstitial macrophages were present in the majority of cases and multifocal lymphocytic myocarditis in a smaller fraction of cases [63,64,65,66,67]. SARS-CoV-2 RNA was detected within the myocardium by multiple techniques and co-localized with rare interstitial cells and cardiomyocytes [64]. The precise identity of the interstitial cells with detectable SARS-CoV-2 RNA is unknown. It is important to note that the majority of autopsy samples are collected after extended periods of hospitalization, beyond the phase of active viral replication. Thus, these analyses likely underestimate the extent of viral infection. 

Pathological analyses have also been performed on patients with clinical diagnoses of myocarditis and heart failure. While these biopsy and autopsy studies have included far fewer subjects, they have each reported macrophage and lymphocyte infiltration, as well as evidence of interstitial cell and cardiomyocyte SARS-CoV-2 infection. Intriguingly, interstitial cells adjacent to areas of microthrombi contained SARS-CoV-2 RNA [68,69,70]. Within the systemic vasculature, endotheliitis with associated viral inclusions and genomes has been observed [71,72]. Collectively, these studies implicate SARS-CoV-2 infection as a possible pathological mechanism contributing to cardiac and vascular manifestations of COVID-19. 

## 3. SARS-CoV-2 Tropism within the Cardiovascular System

In the following sections, we will focus on what is known regarding cardiovascular cell types that are permissive to SARS-CoV-2 infection. We will discuss key findings related to mechanisms of viral entry, replication, propagation, and the consequences of infection. Finally, we will highlight the potential of SARS-CoV-2 virions to trigger local inflammatory responses in the absence of productive infection.

Several studies have established that extrapulmonary sites are susceptible to SARS-CoV-2 infection [73,74,75]. Cellular tropism outside of the lung seems to be dictated by ACE2 expression and the ability of the virus to gain access to extrapulmonary tissues. Whether SARS-CoV-2 enters the heart and vasculature through hematological seeding or immune cell trafficking of virions remains unclear. Among myocardial cell types, cardiomyocytes and pericytes express ACE2 mRNA. Cardiac fibroblasts and vascular smooth muscle cells may also express ACE2, albeit to a lesser degree [76,77]. 

While the detection of SARS-CoV-2 RNA and proteins in autopsy and biopsy specimens is suggestive of viral infection, detailed virologic studies are necessary to demonstrate infectivity. The inoculation of human myocardial cells with SARS-CoV-2 has revealed that cardiomyocytes and pericytes are susceptible to SARS-CoV-2 infection. Intriguingly, endothelial cells, fibroblasts, and macrophages do not support SARS-CoV-2 replication [68,78]. The observations that endothelial cells and macrophages harbor viral RNAs, but are not permissive to SARS-CoV-2 infection, could be explained by the ability of these cell types to interact with and phagocytose viral particles.

The infectious capacity of SARS-CoV-2 towards vasculature remains to be defined. Non-cardiac endothelial cells do not support productive infection [78,79], despite limited evidence of viral RNA in these cells being found on autopsy [72,80]. Intriguingly, human vascular and kidney organoid systems are permissive to SARS-CoV-2, but the targeted cells have not been identified [74]. It remains to be observed whether pericytes and other mural cells within the systemic vasculature support viral replication and contribute to the vascular complications of COVID-19.

## 4. Mechanisms of Cardiomyocyte Infection

Human pluripotent stem cell-derived cardiomyocytes (hPSC-CMs), human cardiac slices, and isolated cardiomyocytes are readily infected by SARS-CoV-2 [68,69,75,81,82,83,84,85,86]. ACE2 serves as the cardiomyocyte cell surface receptor for SARS-CoV-2. Cardiomyocyte infection can be abrogated by either neutralizing ACE2 antibodies or the genetic disruption of *ACE2*. After binding to ACE2, the coronavirus spike proteins must undergo proteolytic activation to initiate membrane fusion [87]. Host proteases located at the plasma membrane (i.e., TMPRSS2) or within endosomes (i.e., cathepsins and calpains) typically perform this function. The relative contributions of each of these protease families to SARS-CoV-2 cell entry vary by cell type [87,88]. hPSC-derived cardiomyocytes robustly express numerous endosomal proteases and low levels of TMPRSS2. Application of the endosomal cysteine protease inhibitor E-64, which blocks cathepsin and calpain activity, abolished SARS-CoV-2 cardiomyocyte infection. The blocking of TMPRSS2 activity, using the serine protease inhibitor camostat mesylate, had no effect on the ability of SARS-CoV-2 to infect cardiomyocytes. Consistent with this result, the introduction of a furin cleavage site mutation (ΔPRRA) into the spike protein of the SARS-CoV-2 strain, which prevents TMPRSS2 cleavage and cytoplasmic cell entry [89,90], also had no effect on cardiomyocyte infectivity. Collectively, these studies demonstrate that SARS-CoV-2 infects cardiomyocytes by binding to ACE2 and entering the cell through an endosomal pathway.

The replication of SARS-CoV-2 within cardiomyocytes requires the viral RNA-dependent RNA polymerase and can be readily inhibited by remdesivir [68,81,91,92,93]. Mature SARS-CoV-2 virions are thought to be released from cardiomyocytes by the exocytosis of lysosomal vesicles [83]. Interestingly, infected cardiomyocytes form syncytia in culture through a spike-dependent mechanism that can be mitigated by furin inhibitors or mutation of the spike furin cleavage site (R682S) [84]. Whether SARS-CoV-2 propagates within the myocardium by directly spreading between adjacent cells through syncytia and bypasses traditional cell entry and exit mechanisms is not yet clear. 

The consequences of SARS-CoV-2 cardiomyocyte infection in vitro include the activation of innate immune pathways, reduced contractility and conduction velocity, and cell death. Infected hPSC-derived cardiomyocytes and human adult cardiomyocytes express pro-inflammatory chemokines (*CCL2*, *CCL7*, *CCL5*, *CCL8*, *CCL11*, *CXCL1*, *CXCL6*, and *CXCL12*) and elicit type I (IFN-α, -β) and type III (IFN-λ) interferon responses [68,69,83,85,86]. The production of CCL2 by infected cardiomyocytes appeared to mediate monocyte chemotaxis. The infection of hamsters, which are naturally permissive to SARS-CoV-2, recapitulated many of these findings throughout the atrial and ventricular myocardium [85]. While IFN-α and IFN-λ pre-treatment protects cardiomyocytes from infection, the endogenous role of IFN signaling following infection is yet to be defined [69]. 

A number of NLRP3 inflammasome-regulated cytokines are elevated in cardiomyocyte infection. However, the role of the inflammasome in cardiomyocyte infection is not well understood. Inflammasome inhibition has been shown to improve functional outcomes in multiple models of cardiac injury [94], and has been proposed to be a target for severe COVID-19 management [95,96]. SARS-CoV-2 proteins can activate the NLRP3 inflammasome in vitro [97,98], and its activation in patients with COVID-19 has also been reported [99,100]. It remains to be observed, however, whether multiple natural and synthetic inflammasome modulators [94,101] can benefit patients with cardiovascular sequalae of COVID-19. 

SARS-CoV-2 infection also leads to marked reductions in cardiomyocyte contractility. Engineered heart tissues composed of hPSC-derived cardiomyocytes and cardiac fibroblasts display reduced force production following infection [68,83]. Cardiomyocyte infection leads to reductions in the expression of genes important for sarcomere function, excitation contraction coupling, and metabolism [68,69,83,85,86]. Immunostaining studies have further revealed evidence of sarcomere breakdown and fragmentation. Each of these findings were evident in autopsy and biopsy samples collected from patients with COVID-19 myocarditis [68,69]. The mechanistic basis of sarcomere breakdown is of considerable interest and remains under investigation. 

Electrophysiological alterations and cardiomyocyte cell death represent late sequelae of cardiomyocyte infection [68,69,75,81,82,83,84,85,86]. Cardiomyocyte cell death has been observed across cardiomyocyte preparations. The inhibition of viral replication was sufficient to prevent cell death and innate immune responses. Conversely, innate immune responses triggered by the sensing of viral nucleic acids failed to impact the extent of cardiomyocyte cell death or sarcomere breakdown [68]. Further studies are necessary to clarify the cell death pathways activated by infection, and to define the mechanistic links between infection, innate immune responses, sarcomere maintenance, and metabolism.

## 5. Mechanisms of Pericyte Infection

Pericytes isolated from the human heart or derived from organoids are also permissive to SARS-CoV-2 infection (Brumback and Dmytrenko, in revision) [102,103]. Cardiac pericytes robustly express ACE2 [104]. Pericytes have a critical role in maintaining endothelial integrity and vascular homeostasis. Conceptually, the infection of pericytes could contribute to the observed vascular manifestations of SARS-CoV-2 infection, including thrombosis, inflammation, and hemodynamic derangements [30,31,32,33,34]. The close proximity between pericytes and vascular endothelial cells may explain why vascular structures within the myocardium contained SARS-CoV-2 RNA [68,69,70,71,72]. Indeed, in situ hybridization and immunostaining for pericyte markers, in addition to SARS-CoV-2 RNAs and proteins in autopsy samples, confirmed that pericytes are a target for SARS-CoV-2.

SARS-CoV-2 readily infects cardiac pericytes within primary cultures and organotypic heart slice preparations (Brumback and Dmytrenko et al., submitted). It is not yet clear if pericytes across tissues and organ systems are all permissive to SARS-CoV-2 infection. It should be noted that cardiac pericytes are unique from an embryologic standpoint, as they are derived from the epicardium [105] and may have properties distinct from pericytes found in other locations. As well as occurring across variants of concern, cardiac pericyte infection is dependent on cell surface ACE2 expression, and proceeds through the endosomal route of entry. The propagation of SARS-CoV-2 between pericytes is not yet understood, but presumably involves viral egress via exocytosis [106]. The role of syncytia formation remains to be addressed. 

The consequences of SARS-CoV-2 pericyte infection include cytokine production, the generation of vasoactive mediators, and cell death. RNA sequencing of infected pericytes revealed marked differential expression of genes associated with the innate immune response to pathogens, type I interferon signaling, leukocyte chemotaxis, and degranulation. Upregulation of vasoactive genes, including endothelin 1 and 2 (EDN1 and EDN2), and downregulation of ACE2, a hallmark of SARS-CoV-2 infection [107], were also observed.

Infected cardiac pericytes remain viable for several days in culture, suggesting that they may serve as a previously unrecognized site of replication and reservoir of the virus. Cardiac pericyte cells did undergo cell death at later stages of infection. The loss of endothelial integrity, endothelial cell dysfunction, basement membrane exposure, and microvascular thrombosis may represent sequelae of cardiac pericyte cell death. Future studies are required to elucidate the pathophysiological consequences of pericyte infection in vivo and their collective contribution to cardiovascular manifestations of COVID-19.

## 6. Non-Infectious Effects of SARS-CoV-2 Virions

SARS-CoV-2 can elicit damaging host responses in various cell types, independent of their permissibility to infection. This phenomenon is referred to as viropathology and implies that viral proteins on the surface of virions act as pathogen-associated molecular patterns (PAMPs). Viropathology differs from intracellular viral nucleic acid sensing as it does not necessarily require cell fusion or viral replication, and, thus, it is not restricted to host cells that express viral entry receptors or replication-competent virions. It is important to note that cells infected by many RNA viruses, including coronaviruses, produce large numbers of replication-incompetent virions that harbor defective viral genomes. SARS-CoV-2 has a particularly high capacity to produce replication-incompetent virions, as a consequence of microhomology-driven genome recombination [108,109]. Replication-incompetent virions have been shown to act as danger signals that induce the production of type I and III IFNs, TNF, IL-6, IL-1β, and other pro-inflammatory cytokines. Consistent with this concept, host inflammatory responses appear to correlate with the emergence of replication-incompetent virions during the later stages of infection [110].

SARS-CoV-2 virions express four structural proteins. Three of these proteins, spike (S), envelope (E), and membrane (M), are expressed on the virion surface; the fourth protein, nucleocapsid (N), is expressed inside the virion and interacts with the viral RNA genome. Purified SARS-CoV-2 S and E proteins induce reactive oxygen species generation, type I IFN, and NFkB responses in multiple cell types, including macrophage cell lines, peritoneal macrophages, alveolar macrophages, and epithelial cells. The N protein may also generate inflammatory responses [98]. Importantly, intratracheal delivery of inactivated SARS-CoV-2 or purified S and E proteins was sufficient to trigger immune cell infiltration into the lung and associated inflammatory gene expression in intact mice [111,112,113]. Type I IFN and NFkB responses induced by E and S require Toll-like receptor 4 (TLR4). Biochemical assays have demonstrated that the S1 subunit of S binds to TLR4 in mouse and human macrophages [114,115].

Our understanding of the potential contributions of viropathology to the pathogenesis of cardiovascular sequelae of COVID-19 is in its infancy. The engulfment of SARS-CoV-2 virions by macrophages and endothelial cells, and the subsequent activation of PRRs, may explain why these cell types contain SARS-CoV-2 RNA in autopsy specimens and display inflammatory phenotypes. The full range of cardiovascular cell types influenced by viropathology is unknown. Intriguingly, the expression of S (S1 subunit) on the surface of cardiomyocytes is sufficient to drive immune cell infiltration, inflammation, and LV remodeling [116]. Future studies in this area are likely to provide key insights pertaining to the etiology of cardiovascular inflammation, myocardial injury, and thrombosis associated with COVID-19.

## 7. Outstanding Questions and Future Directions

As the pandemic has unfolded, scientists and physicians have astutely recognized cardiovascular manifestations of COVID-19 and identified cell types within the heart and vasculature that are susceptible to SARS-CoV-2 infection. Early studies have provided important initial insights into mechanisms of viral cell entry, propagation, and the cellular consequences of infection. However, there remains an enormous amount to learn from both virology and host immune standpoints. The key topics are summarized in Box 1. 

To date, the majority of cardiovascular COVID-19 studies have relied on pathological specimens obtained from infected patients, cell culture, and engineered heart tissue models. Animal models of SARS-CoV-2 infection are critically needed to interrogate host–pathogen responses that contribute to the cardiovascular manifestations of COVID-19. Hamsters are naturally susceptible to SARS-CoV-2 and show evidence of cardiac infection. Viral RNA, inflammatory infiltrates, and microthrombi are present within the heart following the intratracheal inoculation of hamsters with SARS-CoV-2 [85,117]. Mice are permissive to select SARS-CoV-2 variants of concern (B.1.1.7 and B.1.351), and mouse-adapted SARS-CoV-2 viruses have been generated for these purposes [118,119,120]. In addition, transgenic mice that express human *ACE2* either ubiquitously or in select tissues and cell types have been generated and utilized to model SARS-CoV-2 infection [118,121]. The adaption of these reagents to study cardiovascular SARS-CoV-2 infection offers an attractive approach to harness the power of mouse genetics.

Box 1Outstanding questions in cardiac manifestations of COVID-19.
**How does SARS-CoV-2 spread to the heart?** SARS-CoV-2 infection originates in the lung and disseminates to multiple organs in the body, including the heart. It is not clear if cardiac SARS-CoV-2 infection occurs through transient viremia, immune cell trafficking, or by exposure to the virus originating from the pleural and/or pericardial spaces.**Do cardiomyocytes and pericytes express co-receptors for viral entry in addition to ACE2?** Multiple alternate receptors for SARS-CoV-2 have been identified in cell lines of various origins, including CD147 [122], LFA-1 [123], and heparan sulfate [124]. The role of these receptors in cardiac infection remains unknown.**What is the role of cardiomyocyte syncytia formation in clinical disease?** Both viral infection and restricted spike protein expression have been shown to cause syncytia formation between hPSC-derived cardiomyocytes. However, syncytia have rarely been observed in human heart specimens and their contribution to cardiac dysfunction remains to be investigated.**Are there intrinsic defense mechanisms that protect host cardiomyocytes from infection?** While SARS-CoV-2 causes robust infection and cell death of cardiomyocytes in vitro, cardiac infection and injury in human specimens appear more restricted. These observations suggest the existence of restriction factors that may limit viral entry and replication or promote viral clearance and cell survival.**What is the role of direct cardiomyocyte and pericyte infection in acute and post-acute COVID-19 disease progression?** The relative contribution of cardiomyocyte and pericyte infection to cardiac pathology is unclear. Animal models with restricted viral receptor expression may help separate their respective contributions.**Does viropathology contribute to cardiovascular complications of COVID-19?** Macrophages and endothelial cells do not support SARS-CoV-2 replication. However, they may become activated by structural components of SARS-CoV-2 and contribute to organ dysfunction. Immune activation and inflammation without direct viral infection contribute to cardiovascular phenotypes. Animal models will be crucial to dissect this question.**Do variants of concern contribute differently to the progression of cardiac COVID-19?** SARS-CoV-2 variants of concern (VOC) harbor mutations that allow them to be transmitted more easily and potentially increase disease severity. While VOC have been shown to infect cardiac cells, it is not clear if they lead to worsened cardiac pathology or outcomes.**Does vaccination provide protection against long-term cardiac complications after breakthrough infections?** Vaccines have changed the course of the pandemic by dramatically decreasing the transmission and disease severity of COVID-19. However, breakthrough infections still occur in vaccinated individuals. It is unclear if the risks and mechanisms of cardiac complications that follow these breakthrough infections are different from those in unvaccinated cases.


## 8. Concluding Remarks

Cardiovascular involvement in patients with COVID-19 has become increasingly appreciated over the course of the pandemic. Acutely ill individuals present with arrhythmias, myocardial injury, and thromboembolic events. Chronic manifestations include exercise intolerance, chest pain, and fatigue. Recent studies by multiple groups have shown evidence of SARS-CoV-2 RNAs and proteins in the hearts of COVID-19 patients. Cardiomyocytes and cardiac pericytes are permissive to SARS-CoV-2 infection that is dependent on ACE2 and progresses through an endosomal route of cell entry. Infection leads to the release of immune mediators, changes in essential cell function, and the ultimate death of infected cells. Additionally, inflammatory responses elicited by SARS-CoV-2 virions and elaborated cytokines involving a broader array of cardiovascular cell types contribute to cardiac disease. Many key questions remain unanswered and robust animal models are urgently needed to link proposed disease mechanisms to the cardiovascular manifestations of COVID-19 observed in patients.

## Figures and Tables

**Figure 1 viruses-14-01137-f001:**
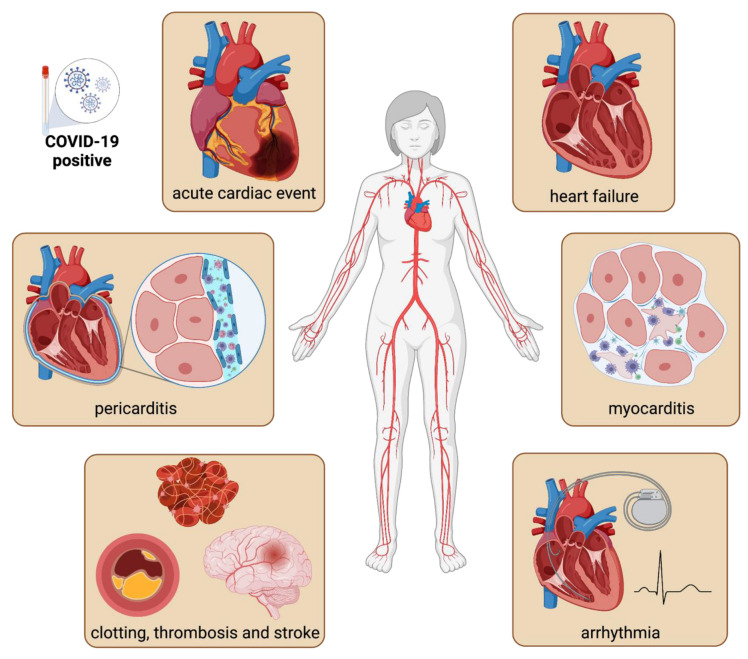
The cardiac complications of COVID-19. Cardiac involvement during the clinical course of COVID-19 can manifest as acute myocardial injury with elevated troponin, heart failure with a decreased ejection fraction, myocarditis, cardiac arrhythmia, thromboembolic events and pericarditis. Evidence of cardiac inflammation can also be present in people who seemingly recover from acute illness.

**Figure 2 viruses-14-01137-f002:**
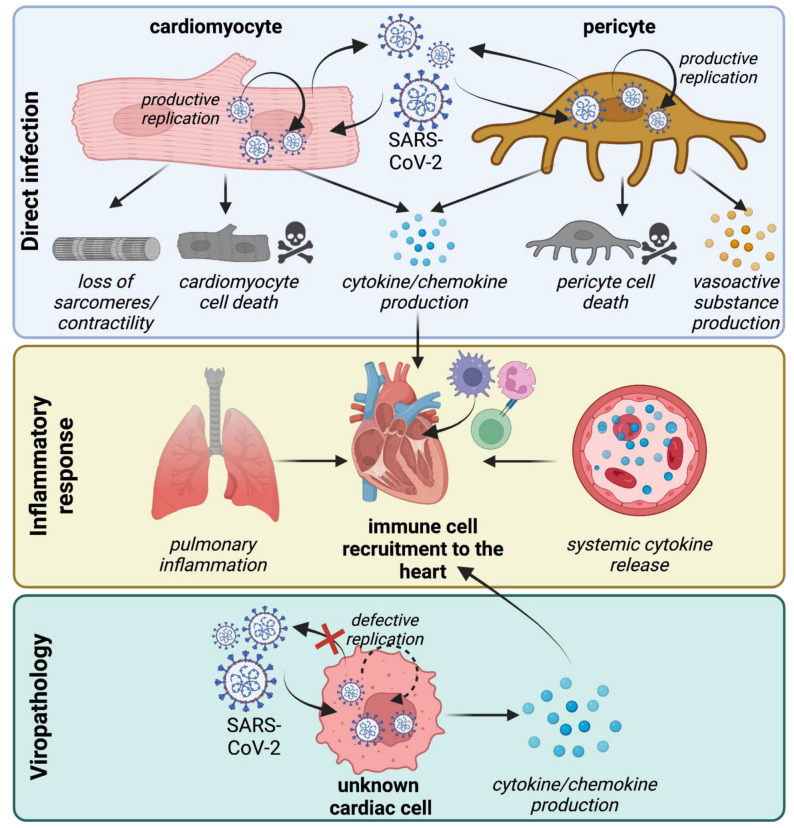
The pathophysiology of cardiac COVID-19. SARS-CoV-2 impacts cardiovascular physiology through the direct infection of cardiac cells, systemic inflammation, and viropathology. Direct infection (**top panel**) has been described in cardiomyocytes and pericytes. Cardiomyocyte infection results in the loss of sarcomere structure, a decrease in contractile force generation, the release of cytokines and chemokines, and the death of infected cells. In pericytes, SARS-CoV-2 infection causes the production of vasoactive substances and cytokines, and the death of infected cells. Productive replication contributes to further viral dissemination. Severe lung pathology in COVID-19 elicits systemic inflammation (**middle panel**) that contributes to immune cell recruitment and a prothrombotic state. SARS-CoV-2 virions act as PAMPs that trigger inflammatory responses in the absence of productive infection. This is referred to as viropathology (**bottom panel**). While poorly understood in the context of the heart, viropathology may regulate the activation of macrophages and endothelial cells, which do not support the replication of SARS-CoV-2. This mechanism can further contribute to immune cell recruitment and the generation of local inflammation within the heart.

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
