# Peer review of "Cardiovascular Tropism and Sequelae of SARS-CoV-2 Infection"

_viruses, 2022, doi:10.3390/v14061137_

Round 1
Reviewer 1 Report
The review performed by Dmytrnko et, is interesting, well written, and presents the most relevant issues about the cardiac tropism of SARS-CoV-2. Some suggestions to improve the article are indicated:
- The title indicates "cardiovascular tropism," but there are no data on the direct vascular effects of the virus on blood vessels. Add information if the virus has direct effects at the vascular level or point out that there is no literature related.
- If possible, add 1 or 2 summary figures of the main ideas of the manuscript. The use of figures facilitates the understanding of the manuscript by readers.
- Please extend the discussion on the epidemiology of cardiac involvement related to infection. Add in references (Xie Y, Xu E, Bowe B, Al-Aly Z. Long-term cardiovascular outcomes of COVID-19. Nat Med. 2022;28(3 ):583-590.doi:10.1038/s41591-022-01689-3) and other articles published in 2022 that have evaluated this.
- Mention the impact of the anti-SARS-CoV-2 vaccination, both its potential to prevent cardiovascular events by lowering the incidence of infection and potential adverse reactions (direct or indirect). Add in references (a.- MMWR report – CDC: https://www.cdc.gov/mmwr/volumes/71/wr/mm7114e1.htm#:~:text=The%20incidence%20of%20cardiac%20outcomes,after%20the% 20second%20vaccine%20dose. b.- Husby A, Hansen JV, Fosbøl E, et al. SARS-CoV-2 vaccination and myocarditis or myopericarditis: population based cohort study. BMJ. 2021;375:e068665. Published 2021 Dec 16. doi: 10.1136/bmj-2021-068665. c.- Bozkurt B, Kamat I, Hotez PJ. Myocarditis With COVID-19 mRNA Vaccines. Circulation. 2021;144(6):471-484. doi:10.1161/CIRCULATIONAHA.121.056135)
Author Response
Reviewer 1
Comments and Suggestions for Authors
The review performed by Dmytrnko et, is interesting, well written, and presents the most relevant issues about the cardiac tropism of SARS-CoV-2. Some suggestions to improve the article are indicated:
- The title indicates "cardiovascular tropism," but there are no data on the direct vascular effects of the virus on blood vessels. Add information if the virus has direct effects at the vascular level or point out that there is no literature related.
We added a paragraph at the end of “SARS-CoV-2 Tropism within Cardiovascular System” to specifically address the lack of evidence for direct endothelial infection. We also discussed the potential for SARS-CoV-2 to infect pericytes (page 6).
- If possible, add 1 or 2 summary figures of the main ideas of the manuscript. The use of figures facilitates the understanding of the manuscript by readers.
We included 2 summary figures of the main cardiovascular effects of SARS-CoV-2 and a Box summarizing open questions and future directions of research for convenience of the reader.
- Please extend the discussion on the epidemiology of cardiac involvement related to infection. Add in references (Xie Y, Xu E, Bowe B, Al-Aly Z. Long-term cardiovascular outcomes of COVID-19. Nat Med. 2022;28(3 ):583-590.doi:10.1038/s41591-022-01689-3) and other articles published in 2022 that have evaluated this.
Additional information and the suggested reference were added to the “Epidemiology of Cardiovascular COVID-19 manifestations” section (page 4).
- Mention the impact of the anti-SARS-CoV-2 vaccination, both its potential to prevent cardiovascular events by lowering the incidence of infection and potential adverse reactions (direct or indirect). Add in references (a.- MMWR report – CDC: https://www.cdc.gov/mmwr/volumes/71/wr/mm7114e1.htm#:~:text=The%20incidence%20of%20cardiac%20outcomes,after%20the%20second%20vaccine%20dose.
b.- Husby A, Hansen JV, Fosbøl E, et al. SARS-CoV-2 vaccination and myocarditis or myopericarditis: population based cohort study. BMJ. 2021;375:e068665. Published 2021 Dec 16. doi: 10.1136/bmj-2021-068665.
c.- Bozkurt B, Kamat I, Hotez PJ. Myocarditis With COVID-19 mRNA Vaccines. Circulation. 2021;144(6):471-484. doi:10.1161/CIRCULATIONAHA.121.056135)
A concluding paragraph with the suggested information was added to the “Epidemiology of Cardiovascular COVID-19 manifestations” section (page 4-5).
Reviewer 2 Report
Manuscript titled "Cardiovascular Tropism and Sequelae of SARS-CoV-2 Infection" describes the cardiovascular complications in patients with COVID-19. Overall structure is of good quality, methodologies and discussion are well performed and references are of good quality but still to be improved,
Authors should improve manuscript in several parts:
1) please add a proper description on the coagulophaties induced by COVID-19
2) a brief chapter should focalized on patients vulnerable like cancer patients following cardiotoxic anticancer therapies ( you can cite 10.3390/cancers12113316)
3)A more appropriate description of the main pathways of cardiotoxicity induced by SARS-CoV-2 should be made, by describing the role of Inflammasome, myddosome and hs-CRP activation in this patients and how selective NLRP3 or myd88 inhibitors should be used in clincial settings ( cite 10.26355/eurrev_202009_22867)
4) considering that some nutraceuticals and medicinal mushrooms exerts pro-immune functions and cardioprotective effects, authors should at least cite he potential role of nutraceuticals as putative cardioprotective strategies in patients with COVID-19 ( please, describe for example Reishi, Cordyceps and others) and how this madicinal mushrooms should be used in patients particularly vulnerable like cancer patients ( cite 10.18632/oncotarget.24984)
Author Response
Reviewer 2
Comments and Suggestions for Authors
Manuscript titled "Cardiovascular Tropism and Sequelae of SARS-CoV-2 Infection" describes the cardiovascular complications in patients with COVID-19. Overall structure is of good quality, methodologies and discussion are well performed and references are of good quality but still to be improved,
Authors should improve manuscript in several parts:
1) please add a proper description on the coagulophaties induced by COVID-19
Thank you for this suggestion. We added a paragraph describing coagulapathies induced by COVID-19, which can be found in the “Epidemiology of Cardiovascular COVID-19 manifestations” section (page 4).
2) a brief chapter should focalized on patients vulnerable like cancer patients following cardiotoxic anticancer therapies ( you can cite 10.3390/cancers12113316)
The discussion of patients at high risk for cardiovascular complications was added to introductory part of the “Epidemiology of Cardiovascular COVID-19 manifestations” section (page 3).
3)A more appropriate description of the main pathways of cardiotoxicity induced by SARS-CoV-2 should be made, by describing the role of Inflammasome, myddosome and hs-CRP activation in this patients and how selective NLRP3 or myd88 inhibitors should be used in clincial settings ( cite 10.26355/eurrev_202009_22867)
Relevant literature on inflammasome pathways was included in the “Mechanisms of Cardiomyocyte Infection” section of the article (page 8).
4) considering that some nutraceuticals and medicinal mushrooms exerts pro-immune functions and cardioprotective effects, authors should at least cite he potential role of nutraceuticals as putative cardioprotective strategies in patients with COVID-19 ( please, describe for example Reishi, Cordyceps and others) and how this madicinal mushrooms should be used in patients particularly vulnerable like cancer patients ( cite 10.18632/oncotarget.24984)
Given the paucity of clinical studies, we did not focus this review on therapeutic strategies. Instead, we decided to add a section on potential immune therapeutic. The suggested reference was added to this section (page 8).